# Glycine Cleavage System and cAMP Receptor Protein Co-Regulate CRISPR/*cas3* Expression to Resist Bacteriophage

**DOI:** 10.3390/v12010090

**Published:** 2020-01-13

**Authors:** Denghui Yang, Zhaofei Wang, Jingjiao Ma, Qiang Fu, Lifei Wu, Hengan Wang, Shaohui Wang, Yaxian Yan, Jianhe Sun

**Affiliations:** 1Shanghai Key Laboratory of Veterinary Biotechnology, Key Laboratory of Urban Agriculture (South), Ministry of Agriculture, School of Agriculture and Biology, Shanghai Jiao Tong University, Shanghai 200240, China; yaonan2345@sjtu.edu.cn (D.Y.); wzfxlzjx@sjtu.edu.cn (Z.W.); majingjiao@sjtu.edu.cn (J.M.); fqiang9@126.com (Q.F.); wlfjy668@sjtu.edu.cn (L.W.); hawang@sjtu.edu.cn (H.W.); 2Shanghai Veterinary Research Institute, Chinese Academy of Agricultural Sciences, Shanghai 200241, China; shwang0827@126.com

**Keywords:** *Escherichia coli*, glycine cleavage system, bacteriophage, cAMP receptor protein, regulate, CRISPR/Cas

## Abstract

The CRISPR/Cas system protects bacteria against bacteriophage and plasmids through a sophisticated mechanism where *cas* operon plays a crucial role consisting of *cse1* and *cas3*. However, comprehensive studies on the regulation of *cas3* operon of the Type I-E CRISPR/Cas system are scarce. Herein, we investigated the regulation of *cas3* in *Escherichia coli*. The mutation in *gcvP* or *crp* reduced the CRISPR/Cas system interference ability and increased bacterial susceptibility to phage, when the *casA* operon of the CRISPR/Cas system was activated. The silence of the glycine cleavage system (GCS) encoded by *gcvTHP* operon reduced *cas3* expression. Adding *N^5^*, *N^10^*-methylene tetrahydrofolate (*N^5^*, *N^10^*-mTHF), which is the product of GCS-catalyzed glycine, was able to activate *cas3* expression. In addition, a cAMP receptor protein (CRP) encoded by *crp* activated *cas3* expression via binding to the *cas3* promoter in response to cAMP concentration. Since *N^5^*, *N^10^*-mTHF provides one-carbon unit for purine, we assumed GCS regulates *cas3* through associating with CRP. It was evident that the mutation of *gcvP* failed to further reduce the *cas3* expression with the *crp* deletion. These results illustrated a novel regulatory pathway which GCS and CRP co-regulate *cas3* of the CRISPR/Cas system and contribute to the defence against invasive genetic elements, where CRP is indispensable for GCS regulation of *cas3* expression.

## 1. Introduction

Prokaryotic viruses occur ubiquitously and pose a serious threat to bacteria and archaea [1]. To resist these extremely large numbers of predators, bacteria have developed multiple resistant mechanisms [2,3,4]. As one of the important defence mechanisms, arrays of clustered regularly interspaced short palindromic repeats (CRISPR) and CRISPR-associated (Cas) proteins constitute the CRISPR/Cas system which has attracted much attention in terms of its function and application in recent years [5,6]. CRISPR arrays are composed of the spacer sequences acquired from foreign DNA between 26 and 72 bp and highly conserved repeat sequences flanked by the spacers [7,8]. A group of *cas* genes encoding Cas proteins is generally located near the CRISPR loci, and it displays immune function associated with CRISPR arrays to battle with foreign DNA invasion [9]. The entire immunization process is divided into three stages: adaptation, expression, and interference. During the first stage, the invasive DNA from phage or plasmids is recognized by Cas proteins. The short fragment of foreign DNA is then integrated into the CRISPR array, becoming a new spacer that functions as a genetic memory. In the second stage, a CRISPR array is transcribed from the leader sequence into a long pre-CRISPR-RNA (pre-crRNA). Pre-crRNA is subsequently processed into small pieces called crRNA, which contain the repeat sequence and the variable spacer derived from the integrated foreign DNA that is complementary to the foreign DNA. In the final stage, the crRNA binds with Cas proteins into a functional complex that can trigger the destruction of invading nucleic acids by base pairing with foreign DNA fragments [10,11].

The status of the CRISPR/Cas system is associated with the biological status of the bacteria. Under normal biological conditions, CRISPR/Cas remains static [12,13]. Furthermore, a continuously activated CRISPR/Cas would constantly integrate new spacers, and the bacterial gene fragment could be mistaken as spacers causing autoimmunity and bacterial death [14]. Therefore, the bacteria activities should be logically regulated, based on which regulation of the function of CRISPR/Cas was explored by the researchers. For instance, *cse1* operon in *Esherichia coli* (*E. coli*) is inhibited by H-NS and activated by LeuO [15]. Furthermore, a recent report has elaborated the relationship between bacteria metabolism and CRISPR/Cas in depth, in which the CRISPR/Cas is promoted by cAMP receptor protein (CRP) and repressed by GalM [16]. CRISPR/Cas systems are grouped into two classes containing six major types by different principles of the effector module design [17]. The CRISPR/Cas of *E. coli* belongs to Type I-E of which the Cas proteins are encoded by the *cas3* operon [*cas3* (*ygcB*)] and the *cse1* operon [*cse1* (*ygcL*), *cse2* (*ygcK*), *cse4* (*ygcJ*), *cas5e* (*ygcI*), *cse3* (*ygcH*), *cas1* (*ygbT*), and *cas2* (*ygbF*)] [15]. However, for the regulation of CRIPSR/Cas in *E. coli*, there were more studies focusing on the *cse1* operon rather than on the *cas3* operon (Figure 1).

The glycine cleavage system (GCS), related to many characters of bacteria, catalyzes the glycine to obtain *N^5^*, *N^10^*-methylene tetrahydrofolate (*N^5^*, *N^10^*-mTHF), which is a one carbon donor for the production of serine, thymidine, and purines. GCS consists of three enzymes and a carrier: *gcvP* (glycine decarboxylase), *gcvH* (lipoic acid-containing carrier), *gcvT* (tetrahydrofolate dependent aminomethyltransferase), and *gcvL* (dihydrolipoamide dehydrogenase) [18]. The CRP is a global regulator that has multiple regulatory effects on bacteria. It performs regulatory functions by forming a CRP-cAMP complex with cAMP and binding to the promoter region of the gene [19]. Our previous work has verified that overexpression or deletion of *cas3* significantly affects bacterial susceptibility to phage infection. Therefore, we used the transposon mutation and DNA pull-down technology to screen the proteins that regulate *cas3* in *E. coli,* and elucidated the mechanisms by which CRISPR/*cas3* is regulated. Our study suggested that GCS affected the bacterial susceptibility to phage by altering *cas3* expression, and CRP was dispensable for the GCS to regulate *cas3* expression.

## 2. Materials and Methods

### 2.1. Strains, Plasmids, and Growth Conditions

The strains, plasmids, and oligonucleotides used in this study are shown in Appendix A. The *E. coli* K-12 strain MG1655 and its derivatives were cultured at 37 °C in Luria-Bertani (LB) or minimal media containing 48 mM Na_2_HPO_4_, 22 mM KH_2_PO_4_, 9 mM NaCl, 19 mM NH_4_Cl, 2 mM MgSO_4_, 100 μM CaCl_2_, and 0.5% (*w/v*) glucose in a shaker at 200 rpm. When necessary, different amounts of nalidixic acid, chloramphenicol (Cm), ampicillin (Amp), glycine, serine, *N^5^*, *N^10^*-mTHF, glucose, or cAMP were added into the media. Bacterial concentration was measured in a SHIMADZU UV-1800 Spectrophotometer at OD_600_. At least three replicates were performed for all experiments.

### 2.2. β-Galactosidase Assays

The reporter plasmids containing *cas3*, *gcv*, *cse1*, or *crp* promoter were transferred into MG1655 and its mutants. The method for determining β-galactosidase (β-gal) activity is described previously [20]. The cultures were taken when the OD_600_ was approximately 1.0. A modified procedure of β-gal assay was used in a transposon mutagenesis experiment to determine the β-gal activity of the reporter strain and its mutants. Briefly, 20 μL of each cultured bacterium was pipetted into 96-well plates and mixed with 80 μL of permeabilization solution (100 mM Na_2_HPO_4_, 20 mM KCl, 2 mM MgSO_4_, 0.8 mg/mL hexadecyltrimethylammonium bromide, 0.4 mg/mL sodium deoxycholate, 5.4 μL/mL beta-mercaptoethanol). These samples were incubated at 30 °C for 30 min. Subsequently, 140 μL of substrate solution (60 mM Na_2_HPO_4_, 40 mM NaH_2_PO_4_, 1 mg/mL o-nitrophenyl-β-D-galactoside, 2.7 μL/mL β-mercaptoethanol) was added into each well. After sufficient color had developed, 160 μL of stop solution (1 M Na_2_CO_3_) was added, and duration of reaction time was noted. The OD_420_ of each sample was recorded using a Biotek ELx800 Microplate Reader. The β-gal activity was calculated by the method described by Miller [20].

### 2.3. Construction and Identification of Transposon Mutants

S17-1 λ pir (pUTmini-Tn5) and reporter strain (*E. coli* MG1655Δ*lacZ*Δ*cas3::lacZ*) were cultured to mid-log phase as the donor and recipient, respectively (see Appendix A). The mutants were selected, and the promoter activity of *cas3* (β-gal activity) was measured by the modified procedure of β-gal assay as described previously. The transposon insertion sites in each mutant were identified by genome walking (Genome Walking Kit, Takara, Kusatsu, Japan).

### 2.4. DNA Pull-Down Assays

The biotin labelled PCR primers for amplifying promoter regions of cas3 gene were commercially synthesized (RuiMian, Shanghai, China) (see Appendix A). DNA pull-down assays were performed as described previously [21]. Briefly, the positive biotin labelled sequence (−172 to 0 of *cas3*) was amplified from MG1655 genomic DNA. The DNA sequence was affixed to beads and then incubated with a supernatant of bacterial lysate. Beads were subsequently washed with buffer containing non-specific DNA, and 50 mM NaCl was used to remove non-adhering and low-specificity DNA-binding proteins. Then 100, 300, and 1000 mM NaCl were used to elute specific DNA-binding proteins. The eluted proteins were separated by sodium dodecyl sulfate-polyacrylamide gel electrophoresis (SDS-PAGE), followed by Brilliant Coomassie staining, and were then analyzed by mass spectrometry.

### 2.5. Western Blotting

Bacteria were cultured to mid-log phase, and 1 mL of bacteria was collected. Polyclonal anti-Cas3 and monoclonal anti-GroEL (Abcam, Cambridge, US) were used for subsequent immunodetection as primary antibodies (see Appendix A). The samples were detected in Chemidoc equipment (BioRad, Hercules, US), and the relative greyscales of Cas3 in different strains were analyzed by software ImageJ (1.4.3.67).

### 2.6. Electrophoretic Mobility Shift Assays

To determine the binding specificity of CRP, the positive probe of the *cas3* gene from MG1655 genomic DNA and the negative probe were cloned into pMD-18T (Takara, Kusatsu, Japan), respectively. The negative probe was from the mutated DNA fragment which the predicted CRP-binding site TATGAGCAGCATCGAA was altered to CTGTGGCAGCAGACTA. CRP was purified according to standard protein purification procedure with slight modification as previously reported [22]. The FAM labelled PCR primers for amplifying promoter regions of the *cas3* gene were commercially synthesized (RuiMian, Shanghai, China). The reaction mixture containing binding buffer (TOLO Biotech, Hefei, China), 10 nM probe, 3 mM cAMP, and different amounts of CRP (0, 100, 200, and 400 nM) was incubated at 26 °C for 30 min. The samples were separated by 2.0% TBE agarose gel. The probes were detected by the ImageQuant LAS 4000 mini (GE Healthcare, Marlborough, US).

### 2.7. DNase I Footprinting Assay

DNase I footprinting assays were performed to identify the CRP-binding sequence described by Wang et al. [23]. Briefly, each probe was prepared with the same method as described in EMSA. Then, the probe (300 ng) was incubated with different amounts of recombinant CRP protein in a total volume of 40 µL. Furthermore, digestion, electrophoresis, and data analysis were carried out with the same procedure described by Wang et al. [23].

### 2.8. Lytic Infection Efficiency Assays

The susceptibility of MG1655 and its mutants to phage was confirmed by PFU assay with certain modifications [24,25]. In brief, strains either containing anti-phage plasmids pGEX3 (pGEX with phage vB_EcoS_SH2 spacer) or not were cultured in LB to mid-log phase. Bacteria were centrifuged at 6000× *g* for 5 min and then resuspended in 10 mM MgSO_4_. For lytic infection, each group of bacteria of approximately 2 × 10^8^ CFU was mixed with 2 × 10^7^ PFU phage, and incubated at 37 °C for 2 h. The mixture was pelleted at 12,000× *g* for 2 min. The supernatants were filtered followed by 10-time serially dilution to evaluate the phage titres on indicator bacterium MC1061.

### 2.9. Plasmid Transformation Assays

The competent state of MG1655 and its mutants at mid-log phase was prepared for chemical transformation. Competent cells were mixed with 500 ng plasmids pGEX, pGEX1 (pGEX with CRISPR1 loci of MG1655 and PAM), or pGEX2 (pGEX with CRISPR2 loci of MG1655 and PAM), respectively, cooled on ice for 30 min, and then heat-shocked followed by recovery in 1 mL LB for 3 h. After serial dilution, the mixture was plated on LBA (100 μg mL^−1^ Amp) and incubated at 37 °C for 16 h. The colonies were counted to determine the plasmid’s transformation efficiency. The stability of the plasmid was determined by the ratio of the number of the pGEX1 or pGEX2 transformed colonies to the pGEX transformed colonies.

## 3. Results

### 3.1. Mutation of gcvP or gcvT Decreases cas3 Expression to Affect Phage Infection

To screen the regulators of the Type I-E CRISPR/Cas system *cas3*, an *E. coli* reporter strain of which the *cas3* open reading frame was replaced by *lacZ* was used to construct a transposon mutant library (see Appendix A). The *cas3* promoter activities of approximately 3000 mutants were measured by modified β-gal assay. The mutants with relatively high or low changes in galactosidase activity were selected as candidates, and some of which were sequenced including *kdpA*, *waaR*, *atpD*, *gcvP*, and so on. A *gcvP* gene, which encodes a pyridoxal phosphate-containing glycine decarboxylase belonging to the GCS, was identified by genome walking since insertion of the transposon led to decrease in *cas3* promoter activity (Figure 2A). The GCS was encoded by the *gcvTHP* and *gcvL* operon [26]. To confirm the impact of *gcvP* and verify whether the reduced *cas3* activity was due to the disruption of the GCS, *gcvP*, or *gcvT* deletion, mutants and their complemented strains were generated. To confirm the *cas3* promotor activity, a reporter plasmid (pRCL1)-containing *cas3* promoter was then transformed into each mutant. The results showed that the *cas3* promoter activity in both Δ*gcvP* (EC20) and Δ*gcvT* (EC40) mutants reduced to half as high as in wild type (WT) (EC1001) and restored in complemented strains CΔ*gcvP* (EC30) and CΔ*gcvT* (EC50) (Figure 2B). The expression level of *cas3* was further determined by Western blotting, which is similar to the reporter plasmid results (Figure 2C). These data indicated that *cas3* expression was affected by the GCS.

The integrated or engineered CRISPR spacers matching invasive DNA is indispensable for the CRISPR/Cas system to exert its immune function [12,27], and the entire *cse1-cse2-cse4-cas5e-cse3* operon is repressed when *E.coli* is under normal biological conditions that led to the CRISPR/Cas remaining static. To activate the CRISPR/Cas, a Δ*hns* mutant was constructed. To determine the effect of GCS on bacterial susceptibility to phage, a plasmid containing the spacer was constructed to match the DNA of phage vB_EcoS_SH2, and was transformed into WT and its Δ*gcvP*, Δ*cas3*, Δ*hns*, Δ*hns*Δ*gcvP*, Δ*hns*Δ*cas3* mutants followed by phage infection (see Text S1 and Appendix A). The control group was transformed with an empty vector.

The results showed that the mutation of *cas3* resulted in a similar susceptibility to phage with the control group. Compared with WT, the phage titer of Δ*hns* was significantly decreased. In contrast, the phage titer of Δ*hns*Δ*gcvP* was approximately twice as high as that of Δ*hns* (Figure 2D). In the control groups, there was no difference among the mutants, which suggested the mutation of these genes had no effect on the other phage-defending mechanisms. These results demonstrated that GCS affected susceptibility to phage by altering *cas3* expression, and Cas3 was crucial for the CRISPR/Cas system to resist phage infection.

To further confirm the impact of *gcvP* on *cas3* when *cse1* was activated, the activator of *cse1* encoded by *leuO* was overexpressed by transforming the p*leuO* to WT, Δ*gcvP*, and Δ*crp.* The results showed that the phage titre of WT (p*leuO*) was significantly decreased compared with WT. In contrast, the phage titre of p*leuO*Δ*gcvP* was approximately twice as high as that of WT (p*leuO*) (Figure 2E).

### 3.2. The cas3 and gcvTHP Promoter Activities are Promoted by Glycine

Since GCS functions via catalyzing glycine, *cas3* expression of WT and mutant Δ*gcvP* cultured in LB media supplemented with glycine was determined to further investigate whether the down-regulation of *cas3* expression was due to the defect of GCS function. In LB media, the addition of 20, 40, 80, and 100 mM glycine induced *cas3* promoter expression of WT, however, the added glycine had no effect on the Δ*gcvP* groups (Figure 3A). To minimize the effect of components of media, the cells were cultured in minimal media. The results showed that the *cas3* expression was activated by the minimal of 1 mM glycine in WT but it failed to be activated by glycine in Δ*gcvP* (Figure 3B), which indicated the knock-out of *gcvP* led to the silence of GCS that the CCS was unable to regulate *cas3* expression through glycine usage.

It was reported previously that the expression of *gcvTHP* operon can be induced by glycine in *E. coli* as well [28]. Therefore, to further investigate the correlation between *gcvTHP* expression and *cas3* expression, the *gcvTHP* promoter was cloned into a reporter plasmid (pRCL2) and transformed into WT. The supplemented glycine in LB media had no effect on the expression of the *gcvTHP* promoter, but the activity of the *gcvTHP* promoter exhibited a higher level in LB than in minimal media without any added glycine (Figure 3C,D). In the minimal media, *gcvTHP* promoter activity started to increase with 1 mM glycine (Figure 3D). These data demonstrated the supplemented glycine was able to promote *cas3* and *gcvTHP* promoter activity in minimal media, and the amount of added glycine to enhance *cas3* promoter activity was consistent with the amount needed to promote *gcvTHP* promoter activity in minimal media (Figure 3B,D).

### 3.3. N^5^, N^10^-Methylene Tetrahydrofolate Promotes cas3 Promoter Activity

In one-carbon unit metabolic pathways, both serine and glycine can be catalyzed into *N^5^*, *N^10^*-methylene tetrahydrofolate (*N^5^*, *N^10^*-mTHF), the same intermediate metabolite derived from serine hydroxymethyltransferase-catalyzed serine or GCS-catalyzed glycine [29,30]. To clarify whether GCS affected *cas3* expression by *N^5^*, *N^10^*-mTHF, *cas3* promoter activity of Δ*gcvP* cultured in LB media supplemented with serine or minimal media with *N^5^*, *N^10^*-mTHF (Toronto Research Chemicals) was detected using pRCL1, since glycine activated the *cas3* expression. For mutant Δ*gcvP*, the *cas3* promoter can be activated by 40 mM serine in LB media (Figure 4A). As expected, 0.1 mM *N^5^*, *N^10^*-mTHF promoted *cas3* promoter activity in both WT and Δ*gcvP* (Figure 4B). Our results demonstrated that GCS indirectly regulated *cas3* promoter activity, and the *N^5^*, *N^10^*-mTHF derived from GCS-catalyzed glycine was able to promote *cas3* promoter activity.

### 3.4. CRP Positively Regulates the Transcription of cas3 Gene to Affect Phage Infection

To further explore the regulators which directly bind to the *cas3* promoter, DNA pull-down assays were employed. The protein bands presented only in experimental group but not in control group were analyzed by mass spectrometry (MS) (Figure 5A). In the upper protein band, eight potential regulators were identified, and in the lower protein band, 63 potential regulators were identified. Among these proteins, the regulatory effect of TnaA, kdgR, AtpD, and CRP on *cas3* were verified. CRP encoded by *crp* was selected among the proteins identified by MS, because cAMP is the essential molecule for CRP function, and it may be associated with *N^5^*, *N^10^*-mTHF. To investigate whether CRP affects *cas3* expression, a Δ*crp* mutant was constructed and pRCL1 was transformed into the WT and Δ*crp* (EC60). Then, *cas3* promoter activity in WT and Δ*crp* cultured in LB supplemented with glucose was analyzed, since it was reported that glucose can reduce the concentration of intracellular cAMP, which is indispensable for CRP–cAMP complex binding to DNA fragments [31]. The *cas3* promoter expression level in WT was almost 4 times higher than that in Δ*crp*. In WT, the presence of glucose led to an approximately 4 times decrease in *cas3* promoter activity. As expected, *cas3* promoter activity was not affected by glucose in Δ*crp*, which indicated cAMP is required by CRP to regulate *cas3* promoter activity (Figure 5B).

To determine the effects of CRP on bacterial susceptibility to phage, the lytic infection efficiency assay was performed. The phage titre of Δ*hns*Δ*crp* was approximately 3 times of Δ*hns*, and the phage titre of pleuOΔ*crp* was almost 3 times of WT (p*leuO*). In addition, there was no difference between WT and Δ*crp* in the control group, which suggested the mutation of *crp* had no effect on the other phage-defending mechanisms (Figure 5C,D). These results demonstrated that CRP affected bacterial susceptibility to phage by altering *cas3* promoter activity.

### 3.5. CRP Activates cas3 Expression by Binding to Transcriptional Initiation Area

To elucidate the regulatory mechanism of CRP protein on *cas3* expression, electrophoretic mobility shift assays (EMSA) were performed. The sequence of the *cas3* (*E. coli* MG1655 strain) promoter was analyzed. Compared with the previously reported core motif of the CRP-binding site (TGTGAN_6_TCACA) of *E. coli*, the transcriptional initiation area of *cas3* in MG1655 harbors a predicted core binding site (TATGAN_6_TCGAA) [32]. Between those, three varied nucleotides were observed. The putative binding site is located at −51 of the *cas3* open reading frame close to the predicted *cas3* promoter (Figure 6A). According to the sequence of putative binding site, a probe amplified from the promoter of *cas3* and its mutant was applied in EMSA. In the original probe group, at least 400 nM CRP protein was required for the entire shift of the 10 nM probe (Figure 6B). The predicted core binding site mutant probe lost its ability to bind to the CRP–cAMP complex, such that even the 400 nM CRP protein was incapable of shifting the mutant probe (Figure 6C).

To further verify this result and clarify the binding motif of CRP, the DNase I footprinting assay was performed. The binding site of CRP to the *cas3* promoter was identified, which is located from −38 to −51 and from −58 to −66, overlapping the predicted core motif of the CRP-binding site (Figure 6D).

To further investigate the regulation of CRP to *cas3* protomer activity, the predicted core binding site mutant promoter was cloned into reporter plasmid (pRCL5). The activity of the mutant *cas3* promoter was almost 3.5 times lower than the original *cas3* promoter (Figure 6E), which indicated that the CRP-cAMP activates *cas3* expression by binding to the transcriptional initiation area (ATGAGCAGN_7_AAATAGCCCGCTG).

### 3.6. The Effect of GCS on cas3 Expression is Mediated by CRP

GCS catalyzes glycine to CO_2_ and NH_3_ and transfers the methylene to H_4_folate to form *N^5^*, *N^10^*-mTHF [26], which might provide one-carbon unit for synthesizing cAMP required for CRP function. Therefore, GCS might regulate *cas3* expression through CRP protein. To investigate whether the absence of CRP would influence GCS regulation of *cas3* expression, the mutant Δ*crp*Δ*gcvP* and the complemented strain were constructed. The expression of *cas3* was evaluated by reporter plasmid and Western blotting. Notably, compared to WT, *cas3* expression of Δ*crp*Δ*gcvP* and Δ*crp*CΔ*gcvP* were significantly reduced. Nevertheless, they were not any lower or higher than the Δ*crp* (Figure 7). In addition, *cas3* expression restored in the complemented strain CΔ*crp*. These results demonstrated that the regulation of GCS on *cas3* expression is mediated by CRP.

### 3.7. The Sufficient Concentration of cAMP Restores cas3 Promoter Activity Reduced by the Mutation of GCS

To clarify whether *cas3* expression regulated by GCS was due to the concentration of cAMP or the level of *crp* expression, *cas3* and *crp* expression was evaluated with reporter plasmids. The *cas3* promoter activity was detected in WT, Δ*gcvP*, and Δ*crp* cultured in LB with different amounts of cAMP addition. *Crp* expression in WT and Δ*gcvP* was also evaluated by reporter plasmids containing the *crp* promoter. Interestingly, the addition of excess cAMP resulted in a significant increase in *cas3* promoter activity in WT and Δ*gcvP*, whereas *cas3* promoter activity was irresponsive to cAMP in Δ*crp*. Notably, *cas3* promoter activity in both WT and Δ*gcvP* reached the same level, which suggested that a sufficient concentration of cAMP could compensate for the decrease in *cas3* promoter activity caused by *gcvP* deletion (Figure 8A). Moreover, in both WT and Δ*gcvP*, gene *crp* expression remained at the same level (Figure 8B). These results indicated that cAMP restored *cas3* promoter activity reduced by the mutation of GCS, and GCS regulated *cas3* promoter activity not by altering *crp* expression. In addition, we utilized HPLC to determine the cAMP, ATP, ADP, and AMP concentration of WT and Δ*gcvP*. Since the cAMP concentration is too low, it is difficult to distinguish the concentration of cAMP in WT and *gcvP* by the HPLC method. Interestingly, the concentration of ATP in WT and Δ*gcvP* is the same, but the concentration of ADP and AMP in Δ*gcvP* is significantly lower than that in WT (Appendix A). This result further confirmed that *N^5^*, *N^10^*_-_mTHF has a certain correlation with cAMP.

### 3.8. The Activation of cse1 by H-NS Has No Effect on the Regulation of cas3 by GCS and CRP

Previous work has reported that the promoter of the cse1 operon is silenced by heat-stable nucleoid-structuring protein (H-NS) in *E. coli* [15]. To clarify the impact of GCS and CRP on the cse1 and cas3 operons in the absence of hns, the cse1 and cas3 promoter activity was firstly evaluated in WT and mutants. The absence of hns led to a 20 times increase in cse1 expression, which indicated hns deletion activated cse1 operon. With hns mutation, the deletion of gcvP had no effect on cse1 expression, and the deletion of crp only led to slight increase in cse1 expression (Figure 9A). In the Δhns (EC90), cas3 promoter activity was slightly increased compared to WT. However, the expression levels of cas3 in ΔhnsΔgcvP (EC100) and ΔhnsΔcrp (EC110) were significantly lower than that in Δhns (Figure 9B). These results suggested that GCS and CRP regulated cas3 promoter activity rather than cse1 expression when the H-NS was absent.

### 3.9. Mutation of gcvP or crp Increases the Stability of Foreign Plasmids

To further investigate the impact of GCS and CRP on foreign DNA within bacteria, the stability of plasmids was evaluated by plasmid transformation assay. The transformation efficiencies of both pGEX1 and pGEX2 which contain the spacer cloned from MG1655, were remarkably increased in Δ*hns*Δ*gcvP*, Δ*hns*Δ*crp*, and ΔhnsΔcas3 compared with Δ*hns* (Figure 10). This elucidated that GCS and CRP regulated the interference of CRISPR/Cas to defend against invasive foreign DNA. Furthermore, GCS and CRP affect interference of CRISPR/Cas by regulating *cas3* expression.

## 4. Discussion

CRISPR/Cas is one of the important defending mechanisms against bacterial phage infection, however, most research has focused on the regulation of the *cse1* operon in *E. coli* [15,24,33]. Herein, we studied the regulation factors of *cas3*, which were screened by transposon mutation and DNA pull-down technology. Previous reports have shown that GCS is related to many biological properties of bacteria [34,35], however, the further mechanism has not been elucidated. Meanwhile, another regulator CRP was identified of which the mechanism may be associated with GCS. Thus, the GCS and the CRP were selected to clarify their mechanism in regulating *cas3* expression.

Glycine is the substrate catalyzed by *gcvP*, and was reported to activate *gcvTHP* operon [28,35]. Nevertheless, our study indicated that supplemented glycine promoted *gcvTHP* expression in minimal media rather than LB. We assumed that the component of LB is complicated, containing glycine, which leads to high expression level of *gcvTHP* regardless of whether there has been supplemented glycine in LB. Thus, the highly expressed GcvTHP is capable of cleaving sufficient exogenous glycine to promote *cas3* promoter expression. However, there is no glycine in the minimal media which keeps the *gcvTHP* operon expression at lower level. Then, with the addition of glycine, the expression of *gcvTHP* increased followed by elevated *cas3* promoter activity.

The present study showed that the predicted core CRP-binding site in MG1655 shared 7 nucleotides with *E. coli* consensus (TGTAAN_6_TCATG), while a previous report shared the other 7 nucleotides with *E. coli* consensus [24]. The mutated probe of the entire predicted core CRP-binding site was constructed which led to the loss of CRP-binding capacity. However, according to the results of the DNase I footprint of the sequenced *cas3* promoter, there were 4 nucleotides overlapped with the predicted CRP core-binding site. As a result, the mutated 4 nucleotides brought the significant decrease of *cas3* expression. Interestingly, although there are few reports on the systematic study of *cas3* regulation, an earlier study identified *cas3* as a part of the CRP regulon, showing high identity with our sequenced CRP-binding site [36].

Using transcriptome and bioinformatic techniques, CRP is found able to regulate the transcriptional initiation of more than 200 promoters within *E. coli*, and many of their regulated mechanisms have been illustrated [36,37,38,39]. In most cases, CRP positively controls gene expression [40], and in our results, the regulation of CRP on *cas3* was consistent with this report. The positive regulation of CRP on the CRISPR/Cas system was also reported previously in *Thermus thermophiles* of which Type I-E (both *cas3* and *cse1* operons) and Type III-A systems are activated by CRP [40]. However, an opposite discovery showed that CRP negatively regulates Type I-E *cse1* operon expression within *E. coli*. This is mainly due to the CRP repressing *cse1* expression by competing for a binding site with LeuO, which is an activator antagonizing H-NS. Moreover, their results mentioned that CRP has no significant influence on *cas3* expression [24]. This inconsistency may be due to the different growing phases of the bacteria collected. In *Pseudomonas aeruginosa*, the quorum-sensing system was observed as regulating Type I-E *cas3* expression reported by Høyland-Kroghsbo et al. In their report, minimal *cas3* expression is found at low cell density, but at a higher cell density, *cas3* expression is activated. In addition, the mutation in both AI synthase *lasI* and *rhlI* leads to a significant decrease in *cas3* expression [41]. Another study reported that when bacteria grows into the stationary phase, the concentration of cAMP is augmented [42]. Our results demonstrate that given sufficient cAMP, *cas3* expression is elevated. Besides, the QS system in *E. coli* was reported to be repressed by CRP. Therefore, according to the correlation among the CRP, QS, and CRISPR/Cas system, we proposed a pathway of the correlation between cell density and the activation of the CRISPR/Cas system, where the increased cAMP concentration at high cell density would enhance the regulatory capability of CRP, leading to an increase in *cas3* expression.

The intermediate metabolite *N^5^*, *N^10^*-mTHF produced by the cleavage of glycine by GCS can provide one-carbon unit for the synthesis of purine nucleotides, which is synthesized in the de novo biosynthetic pathway. In this pathway, the first generated nucleotide is inosine monophosphate, which contributes to the synthesis of various intermediates, including AMP and GMP [43]. Under the continuous catalysis of kinases, ATP and GTP are generated by AMP and GMP, respectively. Therefore, glycine plays a significant role in the biosynthesis of purines, ATP and GTP [44]. In addition, ATP can be catalyzed to remove a pyrophosphate to form cAMP [45]. Moreover, the regulatory function of CRP can be inhibited by glucose because the inactivation of adenylate cyclase by glucose leads to a lack of cAMP [46]. The inhibition of CRP by glucose was verified in our study which suggested cAMP was essential for CRP to regulate the CRISPR/Cas system. Here comes the question whether there is a functional correlation between GCS and CRP when they are regulating *cas3* expression? To answer this question, the *gcvP* further mutated on the Δ*crp* mutant which had no further impact on *cas3* expression, which suggested the indispensable role of CRP for the GCS regulation of *cas3* expression. More importantly, we found that a sufficient concentration of cAMP could restore the *cas3* promoter activity that was reduced by the mutation of *gcvP*. However, the *crp* expression was not affected by *gcvP*, and the Δ*crp* mutant is irresponsive to cAMP. A previous study reported that GCS was found responsible for ATP generation. In one-carbon metabolism cycle, per mole of glycine is used to generate one mole of ATP, with NAD^+^ and NADP^+^ acting as cofactors [47]. In addition, our present study confirmed that ADP and AMP levels were significantly reduced after *gcvP* gene mutation. In summary, we presumed that the interruption of GCS leads to a decrease in cAMP concentration, which affects the formation of CRP–cAMP, and consequently leads to a decreased *cas3* expression (Figure 11).

In fast growing *E. coli*, the CRISPR/Cas system has not shown to be active in WT, either for immunity or for acquisition of new spacers. A previous work showed that by using an engineered spacer matching phage λ, the bacteria gained weak protection against phage infection. However, depression of the CRISPR/Cas system by mutation of the *hns* gene results in a higher level of protection [48]. The activation of CRISPR/Cas in *E. coli* is closely related to a heat-stable nucleoid-structuring protein (H-NS), which can bind non-specifically to the AT-rich region of dsDNA. Therefore, H-NS can inhibit the function of CRISPR/Cas by binding to the *cas* gene promoter [15,33]. Another transcription activator, LeuO, can relieve the H-NS-mediated repression of CRISPR/Cas by competitively binding to the promoter of the *cas* gene, however, LeuO has to be overexpressed to relieve H-NS-mediated repression and the silence of *leuO* has no impact on *cse1* expression [15]. The influence of *cse1* was ruled out, as *hns* was knocked out, when bacterial susceptibility to phage and the stability of foreign plasmids were studied. Given the result that WT and the mutants without the anti-vB_EcoS_SH2 spacer maintained the same level of bacterial susceptibility to phage, it was confirmed that the changed interference ability induced by GCS and CRP was due to the altered *cas3* expression rather than other mechanisms against phage.

## 5. Conclusions

Bacteria have to meet their metabolic requirements to maintain their physiological state including replication, pathogenesis, and defense. Despite the critical nature of this nutritional interaction, the contribution of many metabolic pathways to defend bacteriophage have not yet been examined. To our knowledge, this study identifies the first contribution of the bacterial GCS for preventing them from phage infection. This contribution is closely linked to the CRISPR/Cas system through interaction with CRP. Our study describes the relationship between bacterial metabolism and phage defense mechanisms from the perspective of GCS and CRP. In prokaryotic and eukaryotic cells, GCS serves conservatively as one-carbon provider linked to the ATP generation [49]. During the serine, one-carbon cycle, glycine synthesis pathway, the glycine can be imported from the environment and utilized to generate ATP while maintaining one-carbon cycle. Under resource shortage situation, the bacteria have to keep a lower proliferation rate to save more energy. However, compared to the resource shortage situation bacteria experienced, the pressure from phage seems extremely dangerous. In conditions of resource abundance, the increased bacterial cell density will suffer the high risk of phage infection. Therefore, CRISPR/Cas system expression is dependent on the QS system in some kinds of reported bacteria. Our study implies that when bacteria encounter resource shortages, the CRISPR/Cas system might be shut down to free up more resources to get the bacteria through the tough time, and bacteria will highly express phage-counter measures during the high cell density phase. Taken together, *cas3* expression was regulated by both GCS and CRP in *E. coli*. The GCS and CRP deletion decreased the expression of *cas3*. The CRP regulates *cas3* expression by directly binding to the *cas3* promoter in response to cAMP concentration and is indispensable for GCS regulation of *cas3* expression. The down-regulation of *cas3* expression consequently resulted in the decreased bacterial resistance to phage infection and interference with foreign DNA. Our results highlighted a clear *cas3* expression regulation pathway, which might provide insight into interaction between bacteria and phage.

## Figures and Tables

**Figure 1 viruses-12-00090-f001:**
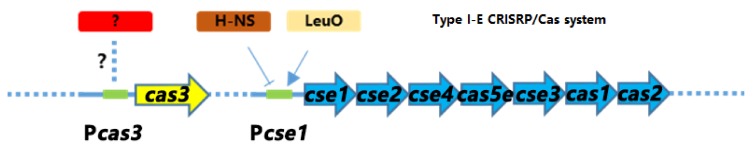
Schematic of the *Esherichia coli* (*E. coli*) Type I-E CRISPR/Cas system. The system consists of *cas3* and *cse1* operon. The *cse1* operon contains 7 genes (blue) consisting of *cse1*, *cse2*, *ces4*, *cas5e*, *cse3*, *cas1*, and *cas2*, with transcription being initiated from the *cse1* promoter. The H-NS represses and LeuO activates *cse1* operon, respectively.

**Figure 2 viruses-12-00090-f002:**
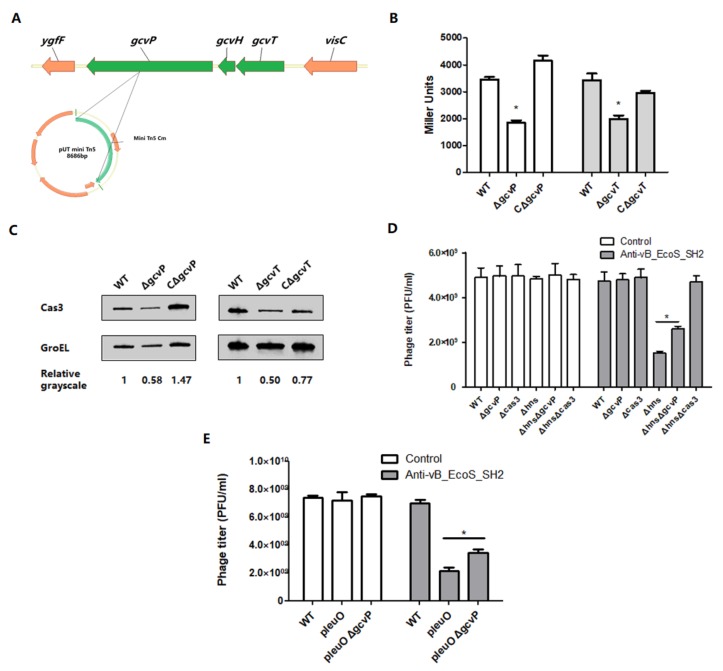
Mutation of *gcvP* or *gcvT* decreases *cas3* expression to affect phage infection. (**A**) The transposon insertion site. (**B**) The β-gal activity of the *cas3* promoter was measured by reporter plasmids in wild type (WT) (EC11), Δ*gcvP* (EC21), CΔ*gcvP* (EC31), Δ*gcvT* (EC41), or CΔ*gcvT* (EC51). The *cas3* promoter activity significantly reduced in Δ*gcvP* and Δ*gcvT*. (**C**) The expression level of *cas3* by Western blotting. The value was calculated from the relative grayscale value of Cas3 to GroEL, and the value of WT was normalized as one. The *cas3* expression significantly reduced in Δ*gcvP* and Δ*gcvT*. (**D**) In the control group, the plasmids without the anti-vB_EcoS_SH2 spacer (pGEX) were transferred into WT (EC15), Δ*gcvP* (EC205), Δ*cas3* (EC135), Δ*hns* (EC95), Δ*hns*Δ*gcvP* (EC105), and Δ*hns*Δ*cas3* (EC125). In the experimental group, the plasmids containing the anti-vB_EcoS_SH2 spacer (pGEX3) were transferred into WT (EC16), Δ*gcvP* (EC206), Δ*cas3* (EC136), Δ*hns* (EC96), Δ*hns*Δ*gcvP* (EC106), and Δ*hns*Δ*cas3* (EC126). The phage titer of Δ*hns*Δ*gcvP* was approximately twice as high as that of Δ*hns*. (**E**) The LeuO overexpression plasmids were transferred into WT and Δ*gcvP*. In the control group, the plasmids without the anti-vB_EcoS_SH2 spacer (pGEX) were transferred into WT (EC141) and Δ*gcvP* (EC142). In the experimental group, the plasmids containing the anti-vB_EcoS_SH2 spacer (pGEX3) were transferred into WT (EC144) and Δ*gcvP* (EC145). The phage titre of p*leuO*Δ*gcvP* was approximately twice as high as that of WT (p*leuO*). Phage titres were measured by lytic infection efficiency assay, as previously described. All the data were mean ± SEM of at least three replicates, and the *p* value (* *p <* 0.05) was analyzed by the *t*-test.

**Figure 3 viruses-12-00090-f003:**
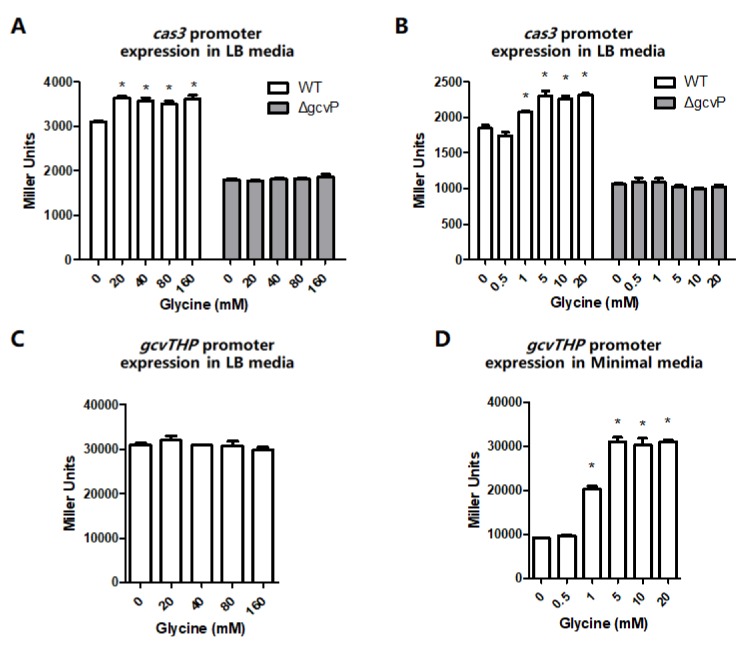
*cas3* and *gcvTHP* promoter expression are activated by glycine. (**A**) The β-gal activity of the *cas3* promoter was measured by reporter plasmids in WT or Δ*gcvP* cultured in Luria-Bertani (LB) media supplemented with 0, 20, 40, 80, or 160 mM glycine. The *cas3* promoter was activated by glycine in WT. (**B**) The β-gal activity of the *cas3* promoter was measured by reporter plasmids in WT or Δ*gcvP* cultured in minimal media supplemented with 0, 0.5, 1, 5, 10, or 20 mM glycine. The *cas3* promoter was activated by the minimal of 1 mM glycine in WT. (**C**) The β-gal activity of the *gcvTHP* promoter was measured by reporter plasmids in WT (EC12) cultured in LB media supplemented with 0, 20, 40, 80, or 160 mM glycine. The supplemented glycine in LB media had no effect on the expression of the *gcvTHP* promoter. (**D**) The β-gal activity of the *gcvTHP* promoter was measured by reporter plasmids in WT cultured in minimal media supplemented with 0, 0.5, 1, 5, 10, or 20 mM glycine. The *gcvTHP* promoter activity started to increase with 1 mM glycine. All the data were mean ± SEM of at least three replicates, and the *p* value (* *p* < 0.05) was analyzed by the *t*-test.

**Figure 4 viruses-12-00090-f004:**
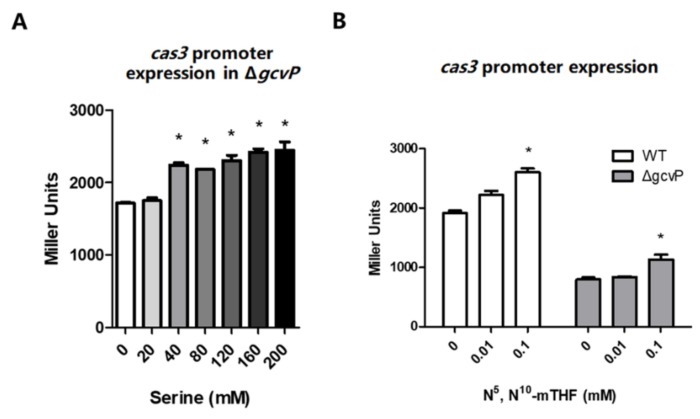
Serine and *N^5^*, *N^10^*-mTHF influence *cas3* promoter activity. (**A**) The β-gal activity of the *cas3* promoter was measured by reporter plasmids in Δ*gcvP* cultured in LB media supplemented with 0, 20, 40, 80, 160, or 200 mM serine. The *cas3* promoter was activated by 40 mM serine in Δ*gcvP* in LB media (**B**) The β-gal activity of the *cas3* promoter was measured by reporter plasmids in WT or Δ*gcvP* cultured in minimal media supplemented with 0, 0.01, or 0.1 mM *N^5^*, *N^10^*-mTHF. 0.1 mM *N^5^*, *N^10^*-mTHF promoted *cas3* promoter activity in both WT and Δ*gcvP.* All the data were the mean ± SEM of at least three replicates, and the *p* value (* *p <* 0.05) was analyzed by the *t*-test.

**Figure 5 viruses-12-00090-f005:**
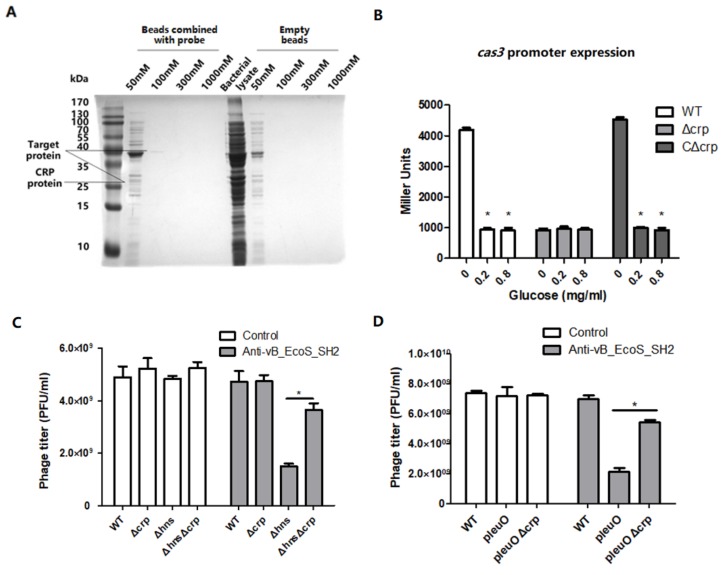
cAMP receptor protein (CRP) positively regulates the transcription of gene *cas3* to affect phage infection. (**A**) The CRP protein was pulled down by magnetic beads bonded to the probe. In the upper protein band, 8 potential regulators were identified, and in the lower protein band, 63 potential regulators were identified. (**B**) The β-gal activity of the *cas3* promoter was measured by reporter plasmids in WT, Δ*crp* (EC61), or CΔ*crp* (EC71) cultured in LB media supplemented with 0, 0.2, or 0.8 mg/mL glucose. In WT, the presence of glucose led to an approximately 4 times decrease in *cas3* promoter activity. (**C**) In the control group, the plasmids without the anti-vB_EcoS_SH2 spacer (pGEX) were transferred into WT (EC15), Δ*crp* (EC605), Δ*hns* (EC95), and Δ*hns*Δ*crp* (EC115). In the experimental group, the plasmids containing the anti-vB_EcoS_SH2 spacer (pGEX3) were transferred into WT (EC16), Δ*crp* (EC606), Δ*hns* (EC96), and Δ*hns*Δ*crp* (EC116). The phage titre of Δ*hns*Δ*crp* was approximately 3 times of Δ*hns.* (**D**) The plasmids overexpressing LeuO were transferred into Δ*crp*. In the control group, the plasmids without the anti-vB_EcoS_SH2 spacer (pGEX) were transferred into Δ*crp* (EC143). In the experimental group, the plasmids containing the anti-vB_EcoS_SH2 spacer (pGEX3) were transferred into Δ*crp* (EC146). The phage titre of pleuOΔ*crp* was almost 3 times of WT (p*leuO*). Phage titres were measured by lytic infection efficiency assay, as described in Materials and Methods. All the data were mean ± SEM of at least three replicates, and the *p* value (* *p* < 0.05) was analyzed by the *t*-test.

**Figure 6 viruses-12-00090-f006:**
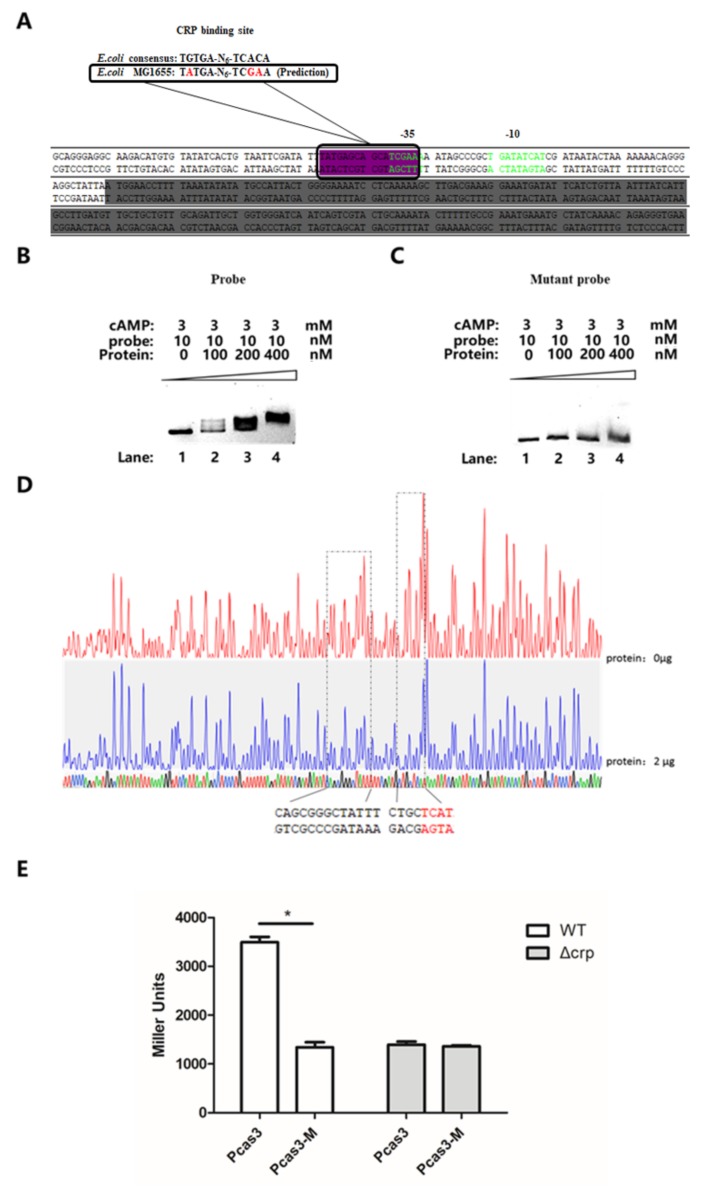
CRP activates *cas3* expression by binding to the transcriptional initiation area. (**A**) The transcriptional initiation area of *cas3*. The green area represents the predicted *cas3* promoter. The purple area represents the putative core binding site of the CRP–cAMP complex. The grey area represents a part of the open reading frame of *cas3*. (**B**) The EMSA for evaluating the binding ability of CRP–cAMP to the putative binding site. In the original probe group, at least 400 nM CRP protein was required for the entire shift of the 10 nM probe. (**C**) The EMSA for evaluating the binding ability of CRP–cAMP to the probe, which was amplified from the mutant sequence of the putative binding site. 400 nM CRP protein was incapable to shift the mutant probe. (**D**) The DNase I footprinting assay for sequencing the exact binding motif of CRP–cAMP. The nucleotides in red were consistent with the predicted binding site. (**E**) The β-gal activity of the original and mutant *cas3* promoter was measured by reporter plasmids in WT (EC19) or Δ*crp* (EC69). The activity of mutant *cas3* promoter was almost 3.5 times lower than the original *cas3* promoter. All the data were mean ± SEM of at least three replicates, and the *p* value (* *p* < 0.05) was analyzed by the *t*-test.

**Figure 7 viruses-12-00090-f007:**
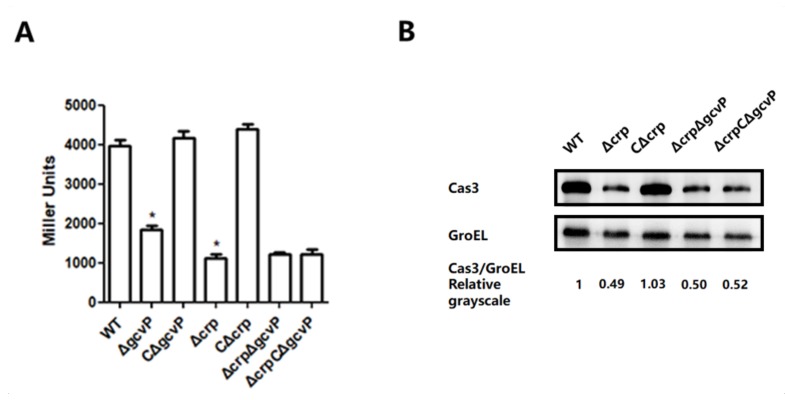
The effect of glycine cleavage system (GCS) on *cas3* expression is mediated by CRP. (**A**) The β-gal activity of the *cas3* promoter was measured by reporter plasmids in WT, Δ*gcvP*, CΔ*gcvp*, Δ*crp*, CΔ*crp* (EC71), Δ*crp*Δ*gcvP* (EC81), or Δ*crp*CΔ*gcvP* (EC151). The *cas3* expression of Δ*crp*Δ*gcvP* and Δ*crp*CΔ*gcvP* were significantly reduced. They were not any lower or higher than the Δ*crp*. (**B**) The expression level of *cas3* by Western blotting. The value was calculated from the relative grayscale value of *cas3* to GroEL, and the value of WT was normalized as one. The result was consistent with β-gal assat. All the data were mean ± SEM of at least three replicates, and the *p* value (* *p* < 0.05) was analyzed by the *t*-test.

**Figure 8 viruses-12-00090-f008:**
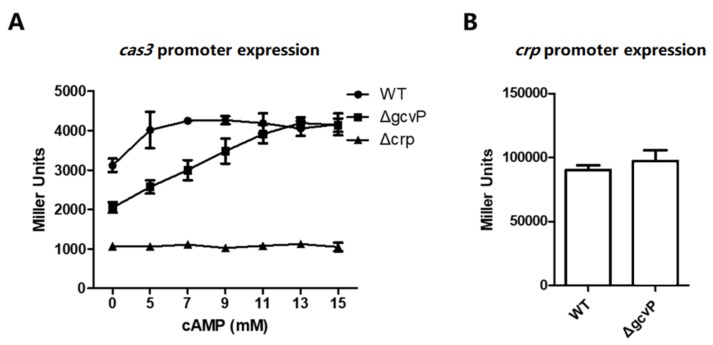
The sufficient concentration of cAMP restores *cas3* promoter activity reduced by the mutation of GCS. (**A**) The β-gal activity of the *cas3* promoter was measured by reporter plasmids in WT, Δ*gcvP*, or Δ*crp* cultured in LB media supplemented with 0, 5, 7, 9, 11, 13, or 15 mM cAMP. The addition of excess cAMP resulted in a significant increase in *cas3* promoter activity in WT and Δ*gcvP*. The *cas3* promoter activity in both WT and Δ*gcvP* reached the same level. (**B**) The β-gal activity of *crp* promoter was measured by reporter plasmids in WT (EC13) or Δ*gcvP* (EC23). In both WT and Δ*gcvP*, gene *crp* expression remained at the same level. All the data were mean ± SEM of at least three replicates, and the *p* value (*p* < 0.05) was analyzed by the *t*-test.

**Figure 9 viruses-12-00090-f009:**
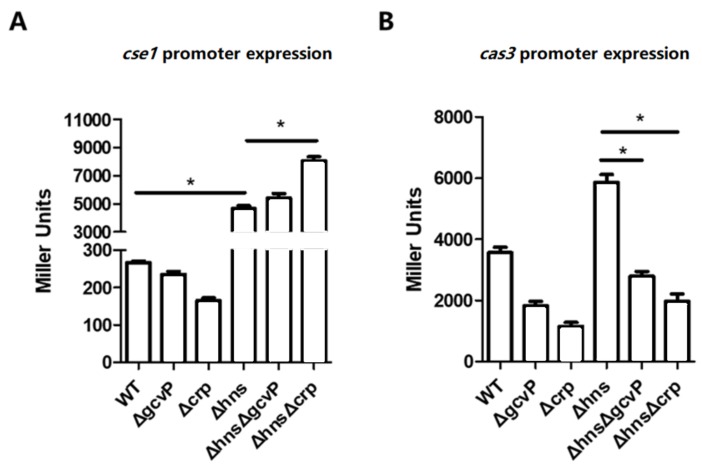
The activation of *cse1* by H-NS has no effect on the regulation of *cas3* by GCS and CRP. (**A**) The β-gal activity of the *cse1* promoter was measured by reporter plasmids (pRCL4) in WT (EC14), Δ*gcvP* (EC24), Δ*crp* (EC64), Δ*hns* (EC94), Δ*hns*Δ*gcvP* (EC104), and Δ*hns*Δ*crp* (EC114) cultured in LB. The absence of *hns* led to a 20 times increase in *cse1* expression, and with *hns* mutation, the deletion of *gcvP* had no effect on *cse1* expression, and the deletion of *crp* led to slight increase in *cse1* expression. (**B**) The β-gal activity of the *cas3* promoter was measured by reporter plasmids in WT, Δ*gcvP*, Δ*crp*, Δ*hns* (EC91), Δ*hns*Δ*gcvP* (EC101), and Δ*hns*Δ*crp* (EC111) cultured in LB. In the Δ*hns*, *cas3* promoter activity was slightly increased compared to WT. The expression levels of *cas3* in ΔhnsΔgcvP (EC100) and Δ*hns*Δ*crp* (EC110) were significantly lower than that in Δ*hns*. All the data were mean ± SEM of at least three replicates, and the *p* value (* *p* < 0.05) was analyzed by the *t*-test.

**Figure 10 viruses-12-00090-f010:**
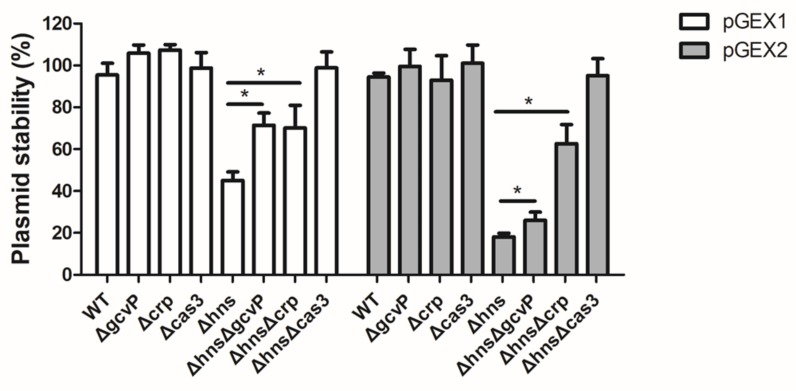
Mutation of *gcvP* or *crp* increases the stability of plasmids. The plasmids containing the CRISPR1 (pGEX1) or CRISPR2 (pGEX2) loci of MG1655 and the empty plasmids (pGEX) were transferred into WT, Δ*hns*, Δ*hns*Δ*gcvP*, Δ*hns*Δ*crp*, and Δ*hns*Δ*cas3*. The stability of the plasmid was determined by calculating the percentage of bacteria containing the pGEX1 or pGEX2 in the bacteria containing the pGEX plasmids. The transformation efficiencies of both pGEX1 and pGEX2, which contain the spacer cloned from MG1655, were remarkably increased in Δ*hns*Δ*gcvP*, Δ*hns*Δ*crp*, and Δ*hns*Δ*cas3* compared with Δ*hns*. All the data were mean ± SEM of at least three replicates, and the *p* value (* *p* < 0.05) was analyzed by the *t*-test.

**Figure 11 viruses-12-00090-f011:**
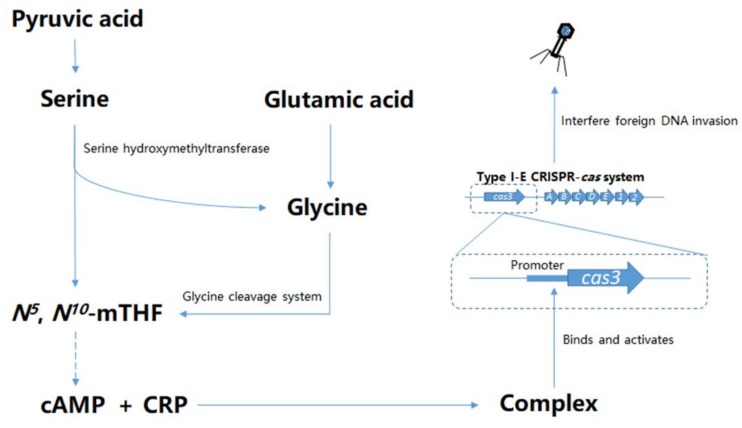
The proposed regulation model of the CRISPR/*cas3* within *E. coli*. The *N^5^*, *N^10^*-mTHF is produced in both serine and glycine metabolism pathways. The disruption of GCS leads to a decrease in *N^5^*, *N^10^*-mTHF. Intracellular cAMP concentration correlates with *N^5^*, *N^10^*-mTHF production. Reduced cAMP weakens the binding ability of the CRP–cAMP complex, which is capable of activating *cas3* expression by binding to the *cas3* promoter. The altered *cas3* expression affects interference with invasive DNA.

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
