# Peer review of "Glycine Cleavage System and cAMP Receptor Protein Co-Regulate CRISPR/cas3 Expression to Resist Bacteriophage"

_viruses, 2020, doi:10.3390/v12010090_

Round 1

Reviewer 1 Report

The manuscript by Yang et al. constitute an investigation of the regulation of expression for Cas3 in the Type I-E CRISPR-Cas system in E. coli.  Cas3 is a critical enzyme in the system responsible for processive target degradation. The study by Yang et al is a contribution to better understanding of its biological function, and complements other studies of regulation by other Cas proteins well.

This reviewer has some key comments that the authors should consider:

A weak point in the manuscript is that the authors argue that GCS system affect regulation of cas3 through Crp. Basically the argument is that as expression level of cas3 is similar in ΔcrpΔgcvP and Δcrp. This is insufficient. To strengthen their case, the authors should consider performing additional experiment with ΔcrpΔgcvP with gcvP complementation. If cas3 expression is not increased in such an experiment, the case for GCS dependence on Crp for Cas3 regulation is stronger. Further, ΔgcvP and CΔgcv results should should be included in Fig 7A for completeness of the experiments and further clarification of the interaction between GcvP and CRP in Cas3 regulation The theoretical basis for a possible GcvP dependence of CRP regulation of Cas3 is proposed to be the fact that GcvP is involved in production of AMP. However, are cAMP levels likely to be affected? Does cAMP synthesis go through AMP? This should be clarified. Line 334-338: In conclusion of the manuscript the authors speculate on the important question of why a phage defense system is regulated by metabolic factors. Their hypothesis is that during resource shortage, CRISPR-Cas need to be shut down to allow the organism to focus resources on cellular maintenance. However, an alternative idea is that in conditions of resource abundance growth cell density will increase which greatly increase the ability for phage to spread. Linking expression of phage defense systems to high metabolic rate may increase chance of survival. The authors should elaborate on hypotheses for why phage defense is linked to metabolism, as they are valuable for understanding the system for directing future research.

Minor comments

Line 64: Should be I-E, not I-F The role of cAMP and CRP in regulation of CRISPR-Cas systems have been investigated in related CRISPR-Cas systems. The authors mention such papers, but should also include the results presented in Patterson et al., Regulation of the Type I-F CRISPR-Cas system by CRP-cAMP and GalM (2015) in the article and discuss their implication. Were additional genes found to be affecting cas gene expression by the transposon mutagenesis? A list of such potential regulators would be valuable to the research community. The authors use an outdated nomenclature for E. coli cas genes. For clarity, the authors should use the nomenclature used by the majority of researchers in the field. Use cse1 instead of casA, cse2 instead of casB, cse4 instead of casC, cas5e instead of casD, and cse3 instead of case. The researchers that introduced the casA-E nomenclature have themselves changed to the new nomenclature. Line 181: Not only cse1 (casA) is repressed but the entire cse1-cse2-cse4-cas5e-cse3 operon. Line 221-222: Lanuguage is unclear, please clarify what you mean. Line 263: The authors should provide a brief and clear summary of the the pull-down assay here. The current description is insufficient. Line 354-355: Was this tested? It seems like a straightforward experiment. Line 442-445: It appears that the authors argue that the correlation between CRISPR-Cas activity and cell density is dependent on increased cAMP level and not quorum sensing in Pseudomonas aeruginosa. The 2017 manuscript by Høyland-Kroghsbo et al. is a firm experimental confirmation of the role of quorum sensing. Please change your statement. The authors repeatedly refer to the expression of lacZ fused to to the Cas3 promoter as “Cas3 expression”, e.g. in fig. 3 and fig. 4. This is not correct, it should be labeled as “Cas3 promoter activity”. Actual Cas3 expression may be affected by other factors than its promoter activity.

Author Response

We have read your review comments very carefully. Thank you for your approval on our research and thank you very much for your review comments so that we can improve our work. According to these comments, we have made some improvements.

Major improvements:

Point 1: A weak point in the manuscript is that the authors argue that GCS system affect regulation of cas3 through Crp. Basically the argument is that as expression level of cas3 is similar in ΔcrpΔgcvP and Δcrp. This is insufficient. To strengthen their case, the authors should consider performing additional experiment with ΔcrpΔgcvP with gcvP complementation. If cas3 expression is not increased in such an experiment, the case for GCS dependence on Crp for Cas3 regulation is stronger. Further, ΔgcvP and CΔgcv results should should be included in Fig 7A for completeness of the experiments and further clarification of the interaction between GcvP and CRP in Cas3 regulation.

Response 1: Thank you for the suggestions. The complemented strain ΔcrpgcvP was constructed followed by the WB and β-gal. The results indicated no significant differences between ΔcrpgcvP and ΔcrpΔgcvP. The results have been added into the manuscript. Meanwhile, ΔgcvP and CΔgcvP results have been added into Fig 7A for the completeness of the experiments. Please find in Fig 7A and 7B.

Point 2: The theoretical basis for a possible GcvP dependence of CRP regulation of Cas3 is proposed to be the fact that GcvP is involved in production of AMP. However, are cAMP levels likely to be affected? Does cAMP synthesis go through AMP? This should be clarified.

Response 2: Thank you for the questions. We have tried different ways to identify the cAMP levels, however the cAMP levels are too low to measure. Thus, we used HPLC to measure AMP levels instead. In the de novo biosynthetic pathway, the first generated nucleotide is inosine monophosphate, which contributes to the synthesis of various intermediates, including AMP and GMP. Under the continuous catalysis of kinases, ATP and GTP are generated by AMP and GMP, respectively. Therefore, glycine plays a significant role in the biosynthesis of purines, ATP and GTP. In addition, ATP can be catalysed to remove a pyrophosphate to form cAMP. The relation between cAMP and AMP had been clarified in Discussion. Please find in line 480-484.

Point 3: Line 334-338: In conclusion of the manuscript the authors speculate on the important question of why a phage defense system is regulated by metabolic factors. Their hypothesis is that during resource shortage, CRISPR-Cas need to be shut down to allow the organism to focus resources on cellular maintenance. However, an alternative idea is that in conditions of resource abundance growth cell density will increase which greatly increase the ability for phage to spread. Linking expression of phage defense systems to high metabolic rate may increase chance of survival. The authors should elaborate on hypotheses for why phage defense is linked to metabolism, as they are valuable for understanding the system for directing future research.

Response 3: Thank you for the comments. In conclusion, we have added some sentences to elaborate on hypotheses for why phage defense is linked to metabolism. “In prokaryotic and eukaryotic cells, GCS serves conservatively as one-carbon provider linked to the ATP generation. During the serine, one-carbon cycle, glycine synthesis pathway, the glycine can be imported from the environment and utilized to generate ATP while maintaining one-carbon cycle. Under resource shortage situation, the bacteria have to keep a lower proliferation rate to save more energy. However, compare to the resource shortage situation bacteria experienced, the pressure from phage seems extremely dangerous. In conditions of resource abundance, the increased bacterial cell density will suffer the high risk of phage infection. Therefore, the CRISPR/Cas system expression is dependent on QS system in some kinds of reported bacteria. Our study implies when bacteria encounter resource shortages, CRISPR/Cas system might be shut down to free up more resources to get the bacteria through tough time, and bacteria will highly express phage-counter measures during high cell density phase”. Please find in 529-540.

Minor improvements:

Point 1: Line 64: Should be I-E, not I-F The role of cAMP and CRP in regulation of CRISPR-Cas systems have been investigated in related CRISPR-Cas systems. The authors mention such papers, but should also include the results presented in Patterson et al., Regulation of the Type I-F CRISPR-Cas system by CRP-cAMP and GalM (2015) in the article and discuss their implication.

Response 1: Thank you for the comment. We have changed “I-F” into “I-E”. And we have cited and discussed the paper of Patterson et al. Please find in lines 57-59.

Point 2: Were additional genes found to be affecting cas gene expression by the transposon mutagenesis? A list of such potential regulators would be valuable to the research community.

Response 2: Thank you for the comment. A simple list of potential regulators identified by transposon has been offered. “The mutants with relatively high or low changes in galactosidase activity were selected as candidates, and some of which were sequenced including kdpA, waaR, atpD, gcvP and so on.” Please find in lines 169-171.

Point 3: The authors use an outdated nomenclature for E. coli cas genes. For clarity, the authors should use the nomenclature used by the majority of researchers in the field. Use cse1 instead of casA, cse2 instead of casB, cse4 instead of casC, cas5e instead of casD, and cse3 instead of case. The researchers that introduced the casA-E nomenclature have themselves changed to the new nomenclature.

Response 3: Thank you for the comment. We have rewritten the nomenclature of CRISPR/Cas proteins. Please find in lines 60-63.

Point 4: Line 181: Not only cse1 (casA) is repressed but the entire cse1-cse2-cse4-cas5e-cse3 operon.

Response 4: Thank you for the comment. We have rewritten this sentence. “The integrated or engineered CRISPR spacers matching invasive DNA is indispensable for the CRISPR/Cas system to exert its immune function, and the entire cse1-cse2-cse4-cas5e-cse3 operon is repressed when E.coli is under normal biological conditions that led to the CRISPR/Cas remains static.” Please find in lines 182-185.

Point 5: Line 221-222: Lanuguage is unclear, please clarify what you mean.

Response 5: Thank you for the comment. We have rewritten this sentence. “The results showed that the cas3 expression was activated by the minimal of 1 mM glycine in WT but it was failed to be activated by glycine in ΔgcvP (Fig 3B), which indicated the knock-out of gcvP led to the silence of GCS that the CCS was unable to regulate cas3 expression through glycine usage”. Please find in lines 226-229.

Point 6: Line 263: The authors should provide a brief and clear summary of the the pull-down assay here. The current description is insufficient.

Response 6: Thank you for the comment. We have written a brief and clear summary of the pull-down assay. “In the upper protein band, eight potential regulators were identified, and in the lower protein band, 63 potential regulators were identified. Among these proteins, the regulatory effect of TnaA, kdgR, AtpD and CRP on cas3 were verified”. Please find in lines 275-277.

Point 7: Line 354-355: Was this tested? It seems like a straightforward experiment.

Response 7: Thank you for the question. The effect of cAMP on cas3 promoter expressions in WT, ΔgcvP and Δcrp had been tested. Initially, in the media containing no cAMP, the cas3 expression was different in WT and mutants. With the increased amount of cAMP, the cas3 expression reached the same level in WT and mutants, which indicated the amount of cAMP is sufficient for CRP expression and the amount of cAMP needed in WT or ΔgcvP were different. This phenomenon was probably due to the different amount of cAMP in WT and ΔgcvP in initial media without supplemented cAMP.

Point 8: Line 442-445: It appears that the authors argue that the correlation between CRISPR-Cas activity and cell density is dependent on increased cAMP level and not quorum sensing in Pseudomonas aeruginosa. The 2017 manuscript by Høyland-Kroghsbo et al. is a firm experimental confirmation of the role of quorum sensing. Please change your statement.

Response 8: Thank you for the comment. We are sorry for puzzling the reviewer with our writing. We did not deny the result by Høyland-Kroghsbo et al. However, the specific mechanism about how QS regulates CRISPR/Cas system has not been elucidated. Some other papers have mentioned that cAMP concentration increases with bacterial cell density and CRP is able to repress the QS system. Therefore, we tried to give more possible explanation about the mechanism on which QS regulating CRISPR/Cas depends. We have discussed the relationship among them in the manuscript. “Besides, the QS system in E.coli was reported to be repressed by CRP. Therefore, according to the correlation among the CRP, QS and CRISPR/Cas system, we proposed a pathway about the correlation between cell density and the activation of the CRISPR/Cas system, where the increased cAMP concentration at high cell density would enhance the regulatory capability of CRP leading to an increase in cas3 expression.” Please find in lines 473-477.

Point 9: The authors repeatedly refer to the expression of lacZ fused to to the Cas3 promoter as “Cas3 expression”, e.g. in fig. 3 and fig. 4. This is not correct, it should be labeled as “Cas3 promoter activity”. Actual Cas3 expression may be affected by other factors than its promoter activity.

Response 9: Thank you for the comment. We have changed all the “cas3 expression” linked to lacZ fusion to “cas3 promoter expression” or “cas3 promoter activity”.

Reviewer 2 Report

    This is a potentially very interesting paper, with a solid experimental design and overall clear conclusions. The authors propose a suggestive model, involving CRP, one of the classic transcriptional regulators in bacteria. However,  in the opinion of this reviewer, several aspects of the manuscript may be improved.   Major points:     1.Regarding experiments with plasmid-harboring strains, I didn't find any information about the induction of the plasmid insert. For instance, in Figure 2, pGEX-derived plasmids are inducible by lactose/IPTG and pBAD with arabinose, did the authors actually check the effect of the inductor in the insert expression?     2.Figure 5: Analyzed protein bands are not clear. I suggest a larger gel and color-coded and more closely arrows. Also, which is the about 38kDa protein? seems highly specifically bound. I am surprised that the authors focused only on the role of CRP as a cas3 regulator, which was already known (ref. 35).     Minor points:   Although I'm not an English native-speaker, I found some odd sentences of the manuscript, too long and with complicated syntaxis. The selection of some verbs and adjectives was not fully appropriate. Page 1, line 33. Bacteriophage should refer only to bacteria viruses, not to archaeal ones. Prokaryotic viruses might be a more suitable term here. Lines 78-80. The role of CRP seems contradictory in these sentences. Please rephrase. Figure legends overall include all the required information. However, figures would benefit from better annotations and/or schemes to clarify each experiment. A larger font is also needed in Figures 1-5. I'd consider splitting some of the complex figures. Line 265: I suggest "analyzed" rather than "sent"  

Author Response

We have read your review comments very carefully. Thank you for your approval on our research and thank you very much for your review comments so that we can improve our work. According to these comments, we have made some improvements.

Point 1: Regarding experiments with plasmid-harboring strains, I didn't find any information about the induction of the plasmid insert. For instance, in Figure 2, pGEX-derived plasmids are inducible by lactose/IPTG and pBAD with arabinose, did the authors actually check the effect of the inductor in the insert expression?

Response 1: Thank you for the suggestions. We are sorry for confuse the reviewer about the background of these plasmids we used. The pGEX and pBAD plasmids were only used for their antibiotic resistance cassette. The DNA fragment representing anti-vB_EcoS_SH2 which was cloned into pGEX contains its own promoter. Therefore, this plasmid is not necessary to be induced. The other plasmid pBAD was constructed by our lab of which only the replication origins and antibiotic resistance cassette are retained. The ORF and its promoter were cloned into this plasmid and it is not an inducible plasmid.

Point 2: Figure 5: Analyzed protein bands are not clear. I suggest a larger gel and color-coded and more closely arrows. Also, which is the about 38kDa protein? seems highly specifically bound. I am surprised that the authors focused only on the role of CRP as a cas3 regulator, which was already known (ref. 35).

Response 2: Thank you for the comments. According to the comments, the gel has been enlarged. We also have written a brief summary of the pull-down assay. “In the upper protein band, eight potential regulators were identified, and in the lower protein band, 63 potential regulators were identified. Among these proteins, the regulatory effect of TnaA, kdgR, AtpD and CRP on cas3 were verified”. However, some of the pulled-down proteins failed to bind to the cas3 promoter. The most abundant 38kDa protein is TnaA, but after EMSA verification, we found the TnaA was unable to bind to cas3 promoter. Although Zheng et al. reported CRP as a potential regulator on cas3 using in vitro and in vivo transcriptional profiling, and the purpose of their research was to identify the CRP regulons. In addition, the mechanism by which the CRP regulates cas3 was not investigated and the cAMP was considered to be linked to bacterial metabolism. Thus, we focused on the role of CRP as a cas3 regulator. Please find in lines 275-277.

Minor improvements:

Point 1: Page 1, line 33. Bacteriophage should refer only to bacteria viruses, not to archaeal ones. Prokaryotic viruses might be a more suitable term here.

Response 1: Thank you for the comment. We have changed “Bacteriophage” into “Prokaryotic viruses”. Please find in line 33.

Point 2: Lines 78-80. The role of CRP seems contradictory in these sentences. Please rephrase.

Response 2: Thank you for the comment. We have rewritten this sentence. “Our study suggested that GCS affected the bacterial susceptibility to phage by altering cas3 expression, and CRP was dispensable for the GCS to regulate cas3 expression.” Please find in lines 80-82.

Point 3: Figure legends overall include all the required information. However, figures would benefit from better annotations and/or schemes to clarify each experiment.

Response 3: Thank you for the comment. More annotation has been added to each figure legend. Please find in figure legends.

Point 4: A larger font is also needed in Figures 1-5. I'd consider splitting some of the complex figures.

Response 4: Thank you for the comment. The figures and the font in Fig 1-5 have been enlarged. Please find in Fig 1-5.

Point 5: Line 265: I suggest "analyzed" rather than "sent"

Response 5: Thank you for the comment. We have changed “sent to” into “analyzed by”. Please find in line 275.

Round 2

Reviewer 1 Report

The authors have responded in an acceptable manner to this reviewers concerns, and performed suggested additional experiments. I have no further s comments on the manuscript